# Sensory Interactions between Sweetness and Fat in a Chocolate Milk Beverage

**DOI:** 10.3390/foods12142711

**Published:** 2023-07-15

**Authors:** Line Pedersen, Anne Sjørup Bertelsen, Derek V. Byrne, Ulla Kidmose

**Affiliations:** 1Food Quality Perception and Society Team, iSense Lab, Department of Food Science, Faculty of Technical Sciences, Aarhus University, 8200 Aarhus N, Denmark; linep@food.au.dk (L.P.); anbei@arlafoods.com (A.S.B.); derekv.byrne@food.au.dk (D.V.B.); 2Sino-Danish College (SDC), University of Chinese Academy of Sciences, Beijing 101408, China

**Keywords:** sugar reduction, fat reduction, acesulfame-K, cross-modal, taste

## Abstract

Fat and sugar-reduced foods and beverages have become increasingly popular for a variety of reasons, mainly relating to health and wellbeing. Depending on the food or beverage, it may be difficult to reduce the fat and/or sugar content and still maintain optimal sensory properties for the specific product. One way of approaching the problem is to gain a better understanding of how a product is affected by a reduction in fat and/or sugar. This paper aims to investigate the sensory interactions between fat and sweetness perception in a chocolate-flavored milk beverage by using a descriptive analysis with a trained sensory panel. The reduction of fat significantly reduced the sweetness intensity of the chocolate milk, while the reduction of sucrose significantly decreased the cream flavor and the fruity and lactic flavor. The perception of acesulfame-K was affected by fat concentration, similarly to sucrose. These results highlight the importance of considering the effects of reducing either sugar and fat on product attributes that are not directly related to the sugar or fat.

## 1. Introduction

Foods and beverages with reduced fat and reduced sugar content have become more and more popular due to an increasing health awareness among consumers and the implementation of laws in some countries that regulate products high in sugar, salt and fat [1,2]. Decreasing the fat and sugar content in food products and beverages would help consumers reduce their calorie intake as well as their intake of excess fat and sugar.

Several countries in Europe are implementing laws and regulations on foods and beverages high in fat, sugar and salt. In the UK, restrictions with respect to product positioning for products high in fat, sugar or salt were recently implemented for online retailers and some physical retailers. Other countries have similar laws and regulations in the pipeline [2]. This has led to a growing demand from the industry and the consumers for fat and sugar-reduced foods and beverages with sensory qualities either directly equivalent to the non-reduced versions or with different sensory qualities which still result in a high liking among consumers.

To be able to produce fat and sugar-reduced products, it is essential to understand the sensory changes in the foods and beverages when their fat and sugar content is reduced. Such sensory changes could be in taste, aroma, texture and/or appearance.

Generally, five basic tastes are acknowledged today: sweet, bitter, sour, salty and umami. However, increasing evidence is pointing at fat as a sixth basic taste [3,4]. The taste of fat is hypothesized to stem from the perception of free fatty acids by the taste receptors CD36 and the G-protein-coupled receptor 120 [5]. However, it may not be the taste of fatty acids that is most interesting in relation to consumer perception and liking, but rather the aromas, textures and flavors associated with the fat.

Sweet taste is detected by the G-protein-coupled taste receptors T1R2 and T1R3, and both nutritive sweeteners (NSs), such as sucrose and fructose, and non-nutritive sweeteners (NNSs), such as acesulfame-K and aspartame, are detected by these receptors [6]. The NSs and NNSs have different sensory properties with respect to, e.g., their levels of sweetness, temporal sweetness profiles, off-flavors and aftertastes [7]. NSs and NNSs also have different stabilities depending on temperature, pH, water activity and storage time [7]. Since NNSs have different sensory and physical properties, their use is very product and process dependent.

Sensory interactions between tastes are called taste–taste interactions. They can be suppressive, such as the bitter taste suppression of sweet taste, and vice versa, at medium and high taste concentrations [8]. Taste–taste interactions may also be enhancing. Thus, medium concentrations of bitter taste may enhance sour taste [8]. Sensory interactions between different modalities such as taste and smell or texture and taste are called cross-modal interactions. Such interactions may also be suppressing as well as enhancing. In relation to fat and sugar reduction, the enhancing properties of other modalities are very relevant. Using aromas to increase the sweetness perception or the perception of fat in foods and beverages is a well-known strategy when reducing either the fat or the NSs without changing the consumers’ perception of the product [9,10,11,12,13,14,15]. Also, tastes, such as sweetness, may enhance the perception of an aroma [13]. This means that removing an ingredient such as sugar or fat may potentially influence not only the sweet and “fatty” taste, but also the perception of other tastes, aromas and textures.

When fat is removed from a food or beverage, it often affects the taste, texture, aroma and/or appearance. This depends on the food or beverage, the amount of fat removed and the starting concentration of fat [16,17]. Many aroma compounds are fat soluble and therefore primarily exist in the fat phase, and when part of the fat is removed, so is part of the fat-soluble aroma compounds [9]. This is often the case when fat native to the food product is removed, e.g., when milk fat is removed from cottage cheese. Increasing the fat concentration may also mask added aroma compounds, especially if they are highly lipophilic [18]. Masked aromas might be added or aromas from other ingredients that are dissolved in fat, such as garlic in a salad dressing [19]. This means that in a food or beverage, added aromas or aromas from other ingredients might become more prominent after a fat reduction [19]. Aromas associated with the fat have been shown to have an effect on the perceived fattiness of milk [10], and addition of aromas has been shown to increase the perceived fattiness in pudding [11] and cheese [12]. The increase in perceived fattiness might be a result of cross-modal interactions, where aromas associated with fatty foods—together with the taste, texture and appearance of the food—may increase the perception of fat [9,11,12]. Thus, to sum up, the literature shows that fat has an important effect on the perception of aromas in a food product, and that aromas may enhance the perception of fat [9,10,11,12]. The aromas naturally present in the fat are enhanced with increasing fat concentration, and aromas added via other ingredients or alone are masked by an increase in fat concentration [9,18].

Removing fat may have a very high impact on the texture and rheology of a food or beverage, depending on the amount of fat in the product and how much is removed. In beverages, such as chocolate-flavored milk, the change in viscosity from the fat reduction is often compensated for by adding hydrocolloids [20]. Addition of hydrocolloids increases the viscosity of the beverage, and this has previously been shown to decrease the sweetness perception of the beverage [21,22,23].

Removing sugar from a food product or beverage will affect the taste [8], aroma [14,15], texture and appearance, as well as the shelf life [24]. Naturally, the sweet taste will be affected by a sugar reduction, but other tastes such as sourness and bitterness will also be affected due to taste–taste interactions. Both sour and bitter taste are enhanced when the sweetness decreases, especially at high concentrations [8]. The reduction of sucrose has been shown to decrease the perception of some aromas, such as vanilla [14].

Often, when sugar is reduced, the sweetness and bulking effect is taken into consideration by adding low-calorie sweeteners and a variety of bulking agents [7]. Many studies have been conducted that tested the physical and sensory effects of different sweeteners and bulking agents in a variety of foods and beverages [25,26,27,28]. For chocolate milk, sweeteners such as stevia, monk fruit, thaumatin, acesulfame-K, aspartame, erythritol, palatinose and sucralose have already been tested either with consumers [29] or using a trained panel [30,31].

Only a few studies have investigated how simultaneous sugar and fat reduction affects the sensory profile and our perception of food and beverages. One study using different biscuits showed that reducing the fat content at different levels, depending on the biscuit, significantly reduced the sweetness perception of the biscuit [17]. Another study using yoghurts revealed that sucrose increased the perceived fattiness, and that fat increased the perceived sweetness [32]. Two other studies showed that fat masked the sweetness in dairy products with varying concentrations of fat [33,34]. Thus, there is some evidence that fat might affect the sweetness of various food products, but it is not clear whether it is a masking or enhancing effect. Also, there is evidence showing that sucrose may affect the perception of attributes related to fat [17,32,33,34].

In this study, we focus on an increased understanding of how simultaneous sugar and fat reduction affects the characteristics of chocolate milk. We chose to work with chocolate milk because chocolate milk has a well-suited matrix where it is possible to regulate both the fat and the sweetness, and it is relatively easy to work with. We also wanted to investigate sugar and fat reduction in a realistic food matrix and, in order to make the results as applicable as possible, to work within the normal range of fat and sugar for chocolate milk. In chocolate milk beverages in Denmark, there is usually between 0.5–3.5% fat and around 5% sucrose. In Denmark, the only sweetener currently used for chocolate milk in the market is acesulfame-K. Thus, we specifically wanted to investigate the effects of acesulfame-K on the sensory attributes of chocolate milk when fat is reduced, and sucrose is substituted with acesulfame-K. We hypothesize firstly, that the concentration of fat will affect the sweetness perception of the chocolate milk; secondly, that the concentration of sucrose will affect attributes related to the “fattiness” of the chocolate-flavored milk; and thirdly, that substitution of sucrose with acesulfame-K in chocolate milk will result in the same sensory profile as chocolate milk sweetened with sucrose.

## 2. Materials and Methods

### 2.1. Study Design and Sample Preparation

We used a full factorial design to test how the main effects of fat and sweetener and the interaction effects between fat and sweetener influenced the sensory perception of chocolate milk. A total of eight samples were tested, i.e., two concentrations of fat (0.1% and 4.5%) in combination with three concentrations of sucrose (0%, 2.5% and 5%) and one concentration of acesulfame-K (0.015% with 2.5% sucrose). An overview of the samples are shown in Table 1. The sweetener concentrations were chosen to represent standard chocolate milk (in the Danish market) with 5% sucrose and an equally sweet, sugar-reduced sample with 2.5% sucrose and acesulfame-K. The samples with 0% sucrose and 2.5% sucrose were chosen to investigate the effect of sucrose on the sensory profile of the chocolate milk. We chose two extreme fat concentrations, compared to what is in the market, to ensure that there was a detectable difference in the fat content between the samples.

We prepared the samples by first making one batch of chocolate milk with 0.1% fat and one batch with 4.5% fat, without any sucrose or acesulfame-K. These batches were prepared by weighing the appropriate amount of skim milk (0.1% fat) (Danmælk^®^, Arla Foods Amba, Viby J, Denmark) and cream (38% fat) (First Price, Arla Foods Amba, Viby J, Denmark), and then mixing the milk and cream with 3% *w*/*w* cocoa powder (Berry Callebaut, Holstebro, Denmark) and 0.035% *w*/*w* kappa carrageenan (Molekymi, Dragør, Denmark). The batches were mixed until the cocoa powder was evenly dispersed. We heated the batches to 85 °C to aid the hydration of the carrageenan and then homogenized them using a TwinPanda 600 (GEA Group, Düsseldorf, Germany). Afterwards, each batch was divided into four samples, and sucrose (Dansukker, København S, Denmark) and acesulfame-K (Sigma-Aldrich, St. Louis, MO, USA) were added as shown in Table 1, providing eight samples in total. We poured 30–35 mL of the samples into 40 mL plastic containers (Corning^®^Gosselin^TM^ TP30C-012, Corning, New York, NY, USA) with a screw lid and stored them at 5 °C until the sensory evaluation (1–3 days later).

### 2.2. Sensory Evaluation

Eleven panelists (nine females, 22–63 years) from the trained sensory panel at the Department of Food Science, Aarhus University evaluated the chocolate milk samples using descriptive analysis. Due to COVID, the panel was throughout the entire process divided into morning and afternoon shifts (five to six panelists each), in order to adhere to social distancing protocols. All panelists consented verbally to participating in the study. The panel received three days of training. The first day featured an introductory discussion generating attributes and a consensus vocabulary on all the samples. Following the discussion, the panel was trained using a subset of the samples on day 2. On day 3, we used PanelCheck (V.1.4.2) (Nofima Mat, Ås, Norway) to monitor panelist performance and discussed the results from the training with the panel. On day 3, we also introduced references for the attributes where there was panel disagreement. The attributes agreed upon after the training can be seen in Table 2. The attributes were evaluated in the following order: aromas, tastes, flavors, textures, after textures and aftertastes, all on a 15 cm line scale with the anchors very low (0 cm) and very high (15 cm) intensity. Flavors were defined as the combined perception of taste and retronasal aroma. There was a 30 s mandatory break after the texture evaluation and a two-minute mandatory break between each sample. The test was conducted over three consecutive days in the sensory booths at the Department of Food Science, Aarhus University. The panelists assessed the samples in triplicate with one repetition each day. Every day, we used a random sample as a warmup. The samples were labelled with a random 3-digit code, and green light was used to mask the color differences between the samples. We served the samples monadically, following a randomized block design. Samples were served at 8°C. The panelists had water, sparkling water and apple slices to clean their palate between samples. Data was collected using EyeQuestion Software (Logic8 BV, Elst, The Netherlands). The local ethical committee for the Central Denmark Region (De Videnskabsetiske Komitéer for region Midtjylland) decided that no ethical approval was required.

### 2.3. Statistical Analysis

To determine the effects of fat and sweetener on the different attributes, we used a mixed model where panelists and replicates were considered as random effects. We created dummy variables for the different levels of fats and sweeteners and considered them as fixed effects. To determine whether there were any significant effects of fat or sweetener we used a three-way analysis of variance (ANOVA). We used Tukey’s highest significant difference multiple comparisons test to determine which levels of fat and sweetener were significantly different. To determine which attributes were significantly affected by the sample, we used mixed models with panelists and replicates as random effects and a sample as a fixed effect in a three-way ANOVA. Mean values for attributes significantly affected by the sample were included in a Principal component analysis (PCA). The statistical analysis was made using XLSTAT version 2020.5.1 (Addinsoft, Paris, France).

## 3. Results

To investigate the effect of fat concentration, sucrose concentration and sugar replacement with acesulfame-K on the sensory perception of chocolate milk, we carried out a sensory descriptive analysis on chocolate milk with two concentrations of fat (0.1% and 4.5%) in combination with three concentrations of sucrose (0%, 2.5% and 5%) and one concentration of acesulfame-K (0.015% with 2.5% sucrose). The results from the ANOVA with the dummy variables fat and sweetener are seen in Table 1, where the attributes with a significant effect of either fat or sweetener are shown.

When the concentration of fat was reduced from 4.5% to 0.1%, the cream aroma was significantly reduced (*p* < 0.0001). The same effect was seen for sweet taste intensity (*p* = 0.037), with a decrease in sweet taste intensity at 0.1% fat compared to 4.5% fat. Both cream flavor (*p* = 0.003) and viscosity (*p* = 0.039) were also significantly reduced when the fat concentration was reduced. For the attributes cocoa flavor (*p* = 0.009), burnt and smoky flavor (*p* = 0.021) and dusty sensation (*p* = 0.003), there was an increase with decreasing fat concentration.

For sweet taste intensity (*p* < 0.0001), chocolate flavor (*p* < 0.0001), cream flavor (*p* = 0.005), fruity and lactic flavor (*p* < 0.0001), viscosity (*p* = 0.017) and mouth-watering (*p* = 0.002), there was a significant increase in these attributes with increasing sucrose concentration. For bitter taste intensity (*p* < 0.0001), cocoa flavor (<0.0001), burnt and smoky flavor (*p* < 0.0001), dusty sensation (*p* < 0.0001), mouth-drying (*p* = 0.004), sour aftertaste (*p* = 0.004) and bitter aftertaste (*p* < 0.0001), there was a significant decrease with increasing sucrose concentration.

The addition of acesulfame-K did not significantly change the sensory profile of the chocolate milk, except for the sweet taste intensity which was significantly higher for the samples with acesulfame-K, as seen in Table 3. The interaction effect between fat and sweetener for the sweet aftertaste was just significant (*p* = 0.05) where the samples without acesulfame-K had a lower sweet aftertaste when the fat concentration was reduced from 4.5% to 0.1% (results not shown). However, the samples with acesulfame-K did not change with respect to sweet aftertaste when the fat concentration was reduced. None of the changes in sweet aftertaste were significant but they might indicate that fat does not affect the temporal sweetness of acesulfame-K in the same way as sucrose.

In Figure 1, PC1 (84.44%) mainly describes the variation in the samples related to sweetness and bitterness, with sweet taste intensity, sweet aftertaste, chocolate flavor and mouth-watering in the lower right-hand corner and bitter taste intensity, bitter aftertaste and mouth-drying in the upper left-hand corner. Samples 4.5F_5S, 4.5F_2.5S_AK, 0.1F_5S and 0.1F_2.5S_AK were evaluated as high in relation to the attributes related to sweetness, while samples 4.5F_2.5S and 0.1F_2.5S were characterized by high bitter taste intensity. PC2 (13.00%) mainly describes variation related to fat concentration, with cream flavor, cream aroma, viscosity and vanilla and caramel aroma at one end (samples 4.5F_0S, 4.5F_2.5S, 4.5F_2.5SAK) and dusty sensation, burnt and smoky aroma, cocoa powder aroma and cocoa powder flavor (samples 0.1F_0S, 0.1F_2.5S, 0.1F_5S, 0.1F_2.5S_AK) at the other end. The samples where part of the sucrose was replaced with acesulfame-K (0.1F_2.5S_AK and 4.5F_2.5S_AK) were placed close to the samples with 5% sucrose (0.1F_5S and 4.5F_5S) in the PCA biplot and were therefore—in accordance with the results from the ANOVA—very similar in their sensory profile. This illustrated that the substitution of 2.5% sucrose with acesulfame-K did not alter the sensory profile of the chocolate milk except for the change in sweetness intensity.

## 4. Discussion

### 4.1. The Effect of Fat Reduction on Sensory Attributes

As seen in Table 3, samples with low fat concentration were perceived as lower in sweet taste intensity when compared to samples with high fat concentration (*p* = 0.037), although not significantly (high fat = 8.9, low fat = 8.5). To our knowledge, no previous studies have examined the effect of fat concentration on the sweetness perception in chocolate milk. One study, using biscuits, found that fat reduction decreased the sweetness perception [17], while a study by Drewnowski and Greenwood (1983) did not find any change in sweetness perception among consumers tasting milk with different levels of fat and sucrose [35]. However, in the study by Drewnowski and Greenwood, they only sampled 16 consumers which is well below the number usually required in a consumer study. In two other studies, they revealed that fat could mask the sweetness in dairy products [33,34]. Since fat does not taste sweet, the reduction in sweet taste intensity, seen in the low-fat samples, might be a result of sensory interactions, such as cross-modal or taste–taste interactions, as increasing evidence is suggesting that “fat” is a taste in itself [3,4,36].

Fat taste might have influenced the perception of sweet taste intensity by suppressing the bitter taste from the cocoa powder. In a review by Khan et al. (2019), they reported two studies showing that the addition of fatty acids to bitter taste solutions suppressed the bitter taste which in turn would no longer suppress the sweet taste [37,38]. In both studies, they also found that the sweetness intensity of sucrose in itself was not affected by the addition of fatty acids [37,38]. Both studies tested the effect of fatty acids in water solutions with only one of the basic tastes. Thus, the bitterness suppression of the effect of fats on the sweetness perception of the same solution has not been investigated. We did not find a sensory interaction effect of fat on the bitterness of the chocolate milk since there were no differences in the bitter taste intensity between the high- and low-fat concentration. Therefore, a decrease in bitterness cannot explain the increased sweetness in the samples with a high-fat concentration compared to the samples with the low-fat concentration. As the addition of fatty acids did not have any effect on the sweet taste intensity itself in the above-mentioned studies, it may not be the taste of fat that is increasing the sweetness perception of the chocolate milks with 4.5% fat compared to 0.1% fat. Instead, the increased sweetness perception might be due to the aroma or viscosity from the milk fat.The perception of cream aroma and viscosity which increased with the higher concentration of fat, might affect the sweetness perception via cross-modal interactions. Cream aroma can, when added to a milk-based food, increase the perception of fat, and perhaps it can also increase the perception of sweetness [9]. Previous studies have investigated the use of aromas to increase the perception of fat in fat-reduced foods via cross-modal interactions [9,10,11,12]. Some have used aromas associated with the type of fat removed, e.g., cream aroma or butter aroma for milk fat [9,12], while others have used an aroma not directly associated with the fat removed, e.g., vanilla aroma in a low-fat pudding [11]. Both methods have been successful in increasing the perception of fat [10,11,12]. Several studies have already shown that adding an aroma that is congruent to a specific sweet food or beverage may significantly increase the sweetness perception [13,15]. As shown in Table 1, there was a significant effect of fat concentration on the perception of cream aroma, where a lower concentration of fat decreased the perception of cream aroma (*p* < 0.001). In many foods, including chocolate milk, the perception of fat is often linked to attributes like viscosity and the ambiguous term “creamy” [33,35].

Earlier studies have shown that viscosity may affect the perception of sucrose and thus sweetness, where an increase in viscosity resulted in a decrease in sweetness [21,22,23], while the results from this study showed that when the viscosity increased in the samples with 4.5% fat compared to the samples with 0.1% fat, this did not seem to decrease the sweetness. The decrease in sweetness perception as a result of the viscosity increase from thickeners has been shown to be an effect of the thickener used and the concentration of the thickener [39]. Therefore, the effect on sweetness from adding thickeners is most likely not a cross-modal interaction, but a physical effect of the thickener itself. Our results did show, however, that increasing the fat concentration and thus the viscosity resulted in a higher level of sweetness (*p* = 0.039). This effect cannot be ascribed to the increase in viscosity alone but should also be considered as an effect of the aroma. Likewise, fat significantly lowered the dusty sensation of the chocolate milk samples (*p* = 0.0003). This might be a result of the increase of viscosity and the lubricating properties associated with fat.

### 4.2. The Effect of Sucrose Reduction on Sensory Attributes

As expected, lower concentrations of sucrose resulted in significantly lower sweet taste intensity (*p* < 0.0001) and significantly higher bitter taste intensity (*p* < 0.0001). Previous studies working with sugar reduction in chocolate milk have found similar results [14,30]. We know that sweetness can mask bitterness, and that these two tastes are often inversely correlated [14]. Similarly, the results from this study showed an inverse correlation between sweetness and bitterness, as seen in the PCA biplot in Figure 1. Lower concentrations of sucrose also resulted in significantly higher levels of sour (*p* = 0.004) and bitter (*p* < 0.0001) aftertaste which probably is an effect of the sucrose masking the sour and bitter taste when present at higher concentrations [8].

As expected, the sucrose concentration did not affect the perception of any of the aroma attributes. However, lowering the concentration of sucrose affected many of the flavor attributes. Chocolate flavor, cream flavor and fruity/lactic flavor were significantly lower when the sucrose concentration was lowered (Table 3). For chocolate flavor, there was a significant reduction from 5% to 2.5% sucrose and again from 2.5% to 0% sucrose, whereas for cream flavor and fruity and lactic flavor we only saw a significant decrease from 2.5% to 0% sucrose. The cream flavor has most likely higher intensity with higher sucrose concentrations because of cross-modal interactions between the sweet taste of sucrose and the aroma and/or mouthfeel of the cream. Previous studies have found similar results when combining sucrose with aromas associated with sweet taste, such as vanilla [13]. It seems reasonable to assume that the aroma and/or mouthfeel of cream by the panelists would be associated with sweetness.

The decrease in chocolate flavor was likely a result of cross-modal interactions between the corresponding aroma and sweetness. The PCA biplot in Figure 1 supports this as chocolate flavor was correlated to sweetness. If the sample design had included varying levels of cocoa powder, then perhaps the chocolate flavor would have been located some place between the sweetness intensity and cocoa powder in the biplot. The same effect for chocolate flavor has been found in previous studies [14,40], where the chocolate flavor in both chocolate-flavored milk and milk chocolate decreased with lower concentrations of sucrose. Cream flavor and fruity and lactic flavor were both represented on PC2 and were therefore not correlated with sweetness even though there was a significant effect of sweetener in the ANOVA.

Cocoa powder flavor and burnt and smoky flavor were both significantly higher in intensity when the sucrose concentration was reduced. The increase in cocoa powder and burnt and smoky flavor might be a result of cross-modal interactions between the corresponding aroma and bitterness. In the PCA biplot in Figure 1, the flavors cocoa powder and burnt and smoky flavor are located right between the bitter taste intensity and the cocoa powder aroma and burnt and smoky aroma. These flavors seem to be equally correlated with bitter taste intensity, cocoa powder and burnt and smoky aroma.

The viscosity (*p* = 0.017) and the mouth-watering (*p* = 0.002) were significantly lower when the sucrose concentration was lowered. Removing sucrose from chocolate milk affects the viscous properties of the milk due to the bulking properties of sucrose, thus making the chocolate milk less viscous, and increasing the dusty sensation which most likely arises from the cocoa powder. As seen in Table 1, the samples with 0% sugar had the lowest viscosity, while the samples with 5% sucrose had the highest viscosity. Both dusty sensation (*p* < 0.0001) and mouth-drying (*p* = 0.004) increased with lower levels of sucrose. Sucrose is known to increase salivation [41] and, thus, the mouth-watering would decrease when sucrose is lowered while the mouth-drying would increase.

### 4.3. The Effect of Acesulfame-K

The results from the post hoc analysis in Table 3 showed that the sample with 2.5% sucrose and acesulfame-K had a higher sweetness than the sample with 5% sucrose, but the bitterness was the same. This was also evident in the PCA biplot (Figure 1). Acesulfame-K is known to have a bitter taste [42], but usually not at lower concentrations [43].

Mixing different sweeteners may give synergistic effects on sweetness, but this was not the case for acesulfame-K and sucrose at concentrations equivalent to 5% sucrose [44]. Thus, the difference in sweetness intensity between the 5% sucrose samples and the samples with 2.5% sucrose and acesulfame-K must be a result of the 0.015% acesulfame-K being sweeter than the 2.5% sucrose which it was replacing. There was a significant decrease in the fruity and lactic flavor from the sample with 2.5% sucrose and acesulfame-K to the sample with 2.5% sucrose, which was not the case from the sample with 5% sucrose to the sample with 2.5% sucrose. However, the sample with 2.5% sucrose and acesulfame-K was not significantly different from the sample with 5% sucrose.

The viscosity of the samples with 2.5% sucrose and 2.5% sucrose with acesulfame-K was not significantly different, while the samples with 2.5% sucrose had a significantly lower viscosity than the samples with 5% sucrose. This was not the case for the samples with 2.5% sucrose and acesulfame-K. The amount of acesulfame-K added (0.015% *w*/*w*) was not expected to have a bulking effect. Hence, the increase in viscosity from acesulfame-K must be derived from something else.

Acesulfame-K was able to maintain the levels of all the attributes affected by sucrose, except for sweetness itself, where the sample with acesulfame-K was rated as significantly sweeter than the 5% sucrose samples. Acesulfame-K could therefore be a good substitute for sucrose, at least as a partial sugar reduction in chocolate milk, irrespective of the level of fat.

### 4.4. Comparison of Fat and Sucrose-Reduction Effects

Reducing fat affects less attributes compared to reducing sucrose, where taste, flavor, texture, aftertaste and aftertexture attributes were affected. Even though fat reduction affects less attributes compared to sucrose reduction, the impact on the consumer liking of the product may be just as high because attributes can have different levels of importance for liking. For chocolate milk, it has previously been shown that sweetness is an important driver of liking [45], while for chocolate, both sweetness, fat and melting properties play a very important role for liking [46]. For vanilla ice cream, the sweetness, white chocolate and powdered milk aroma were more important for consumer acceptance than the texture attributes creaminess and spreadability, when comparing three different ice creams with their fat and sugar-reduced versions [47].

In this study, sucrose and fat did not only affect attributes directly associated with their own aroma, taste, flavor and texture qualities. They both affected attributes related to the other, e.g., sweet taste intensity was reduced by fat reduction and cream flavor was reduced by reduction of sweetener. Thus, when reducing fat and sucrose in a food product or beverage, this study indicates that it is important to consider the effect on attributes not directly associated with the ingredient being reduced.

### 4.5. Limitations and Future Considerations

Optimally, the training should have run in one shift to help ensure overall panel agreement on the attributes. However, the panel is used to working in two shifts and was therefore experienced when it came to understanding attributes and agreeing on attributes between shifts. Some of the panelists were relatively new to the panel, and some did not have any prior experience as sensory panelists. Panel alignment and replicability was checked in PanelCheck (V.1.4.2) (Nofima Mat, Ås, Norway). There was no attribute disagreement between the two shifts, and none of the panelists had issues with replicability or discriminability.

The fat reduction in this chocolate milk was relatively high for chocolate milk (4.5% to 0.1%); however, compared to other high fat and high sugar products such as chocolate or pudding, the fat reduction was relatively small. It would be interesting to investigate the effects of both a low and a high-fat reduction in one of the products, to see how removal of fat affects the product characteristics. The same goes for higher concentrations of sugar. There was no sample that was sweetened exclusively with acesulfame-K which might otherwise have been very interesting because the results indicate that the concentration of fat might affect acesulfame-K differently when it comes to sweet aftertaste, and maybe other attributes.

Except for the fruity and lactic flavor and dustiness, we did not see an effect of replicate for any of the attributes. The samples might have become less fruity and lactic in flavor as a result of fat coalescence after the homogenization which may also have reduced the dusty sensation of the chocolate milk. These are, however, just speculations. A longitudinal study of the changes in sensory and physical characteristics of chocolate milk after homogenization has to be conducted in order to properly evaluate if the changes were an effect of time and fat coalescence.

## 5. Conclusions

Firstly, we hypothesized that fat would affect the sweetness perception of the chocolate milk samples, and it did. Reducing the fat concentration significantly reduced the sweetness intensity of the chocolate milk. Secondly, we hypothesized that the concentration of sucrose would also affect attributes related to the milk fat. The results showed that reducing the sucrose concentration did affect the cream flavor and the fruity and lactic flavor as a reduction in sucrose decreased the perception of these two flavors. Lastly, we hypothesized that acesulfame-K would have the same sensory effects as sucrose, which it did, except for the attribute sweet taste intensity where the samples with acesulfame-K had a significantly higher sweet taste intensity.

When changing the amount of either fat and sucrose, or both at the same time, it is important to not only consider the direct effects of the change, e.g., a change in sucrose resulting in a change in sweetness, but also the indirect effects. Thus, to make sugar and fat-reduced food products with a high consumer liking, it is important to consider how the product as a whole is affected by a reduction in fat and sugar.

## Figures and Tables

**Figure 1 foods-12-02711-f001:**
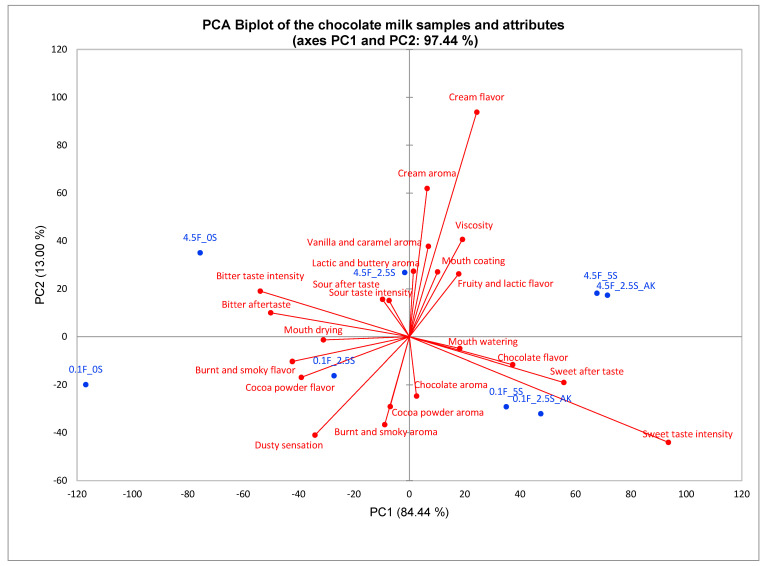
Biplot from the principal component analysis of the attributes where there was a significant effect of the samples in the three-way ANOVA. Samples are blue and attributes are red. 4.5F_0S: 4.5% fat and 0% sucrose; 4.5F_2.5S: 4.5% fat and 2.5% sucrose; 4.5F_5S: 4.5% fat and 5% sucrose; 4.5F_2.5S_AK: 4.5% fat, 2.5% sucrose and 0.015% acesulfame-K; 0.1F_0S: 0.1% fat and 0% sucrose; 0.1F_2.5S: 0.1% fat and 2.5% sucrose; 0.1F_5S: 0.1% fat and 5% sucrose; 0.1F_2.5S_AK: 0.1% fat, 2.5% sucrose and 0.015% acesulfame-K.

**Table 1 foods-12-02711-t001:** The sample compositions of all the included chocolate milk samples. 0.1F_0S: 0.1% fat and 0% sucrose; 0.1F_2.5S: 0.1% fat and 2.5% sucrose; 0.1F_5S: 0.1% fat and 5% sucrose; 0.1F_2.5S_AK: 0.1% fat, 2.5% sucrose and 0.015% acesulfame-K; 4.5F_0S: 4.5% fat and 0% sucrose; 4.5F_2.5S: 4.5% fat and 2.5% sucrose; 4.5F_5S: 4.5% fat and 5% sucrose; 4.5F_2.5S_AK: 4.5% fat, 2.5% sucrose and 0.015% acesulfame-K. *w*/*w*: weight/weight.

Sample Name	Milk Fat (% *w*/*w*)	Sucrose (% *w*/*w*)	Acesulfame-K (% *w*/*w*)
0.1F_0S	0.1	0	-
0.1F_2.5S	0.1	2.5	-
0.1F_5S	0.1	5	-
0.1F_2.5S_AK	0.1	2.5	0.015
4.5F_0S	4.5	0	-
4.5F_2.5S	4.5	2.5	-
4.5F_5S	4.5	5	-
4.5F_2.5S_AK	4.5	2.5	0.015

**Table 2 foods-12-02711-t002:** Lists all sensory attributes, evaluated in the descriptive analysis, their description and references (if references were used).

Attribute	Panel Definition	Reference
Aroma		
Overall aroma	The overall aroma intensity of the sample	Panel definition
Cocoa powder aroma	The earthy and cocoa powder notes of the sample	Cocoa powder (Berry Callebaut, Holstebro, Denmark)
Dark chocolate aroma	The aroma of dark chocolate and/or dark and heavy fruits like plums	Panel definition
Cream aroma	The aroma of milk cream from cows and light fruits like peaches and fresh apples	Fresh cream with 38% fat (First Price, Arla Foods Amba, Viby J, Denmark)
Lactic and buttery aroma	The aroma of butter and lactic acid	Salted butter (Engvang Lidl, Neckarsulm, Germany)
Burnt and smoky aroma	A burnt, coffee-like, smoky aroma	Panel definition
Vanilla and caramel aroma	Aroma of vanilla and caramel might be associated with milk chocolate	Panel definition
Off odor	Animalic and sulfuric smell	Panel definition
Taste		
Sweetness	The sweet taste intensity	Panel definition
Bitterness	The bitter taste intensity	Panel definition
Sourness	The sour taste intensity	Panel definition
Flavor		
Cocoa flavor	The flavor of cocoa powder and dusty/earthy notes	Panel definition
Chocolate flavor	The flavor of chocolate	Panel definition
Cream flavor	The flavor of milk cream from cows	Fresh cream with 38% fat (First Price, Arla Foods Amba, Viby J, Denmark)
Fruity and lactic flavor	The flavor of very ripe fruit and lactic acid	Lactic acid on smelling strips
Burnt and smoky flavor	Coffee-like, smoky and roasted flavors	Panel definition
Off flavor	Fermented and sulfuric flavors	Panel definition
Texture		
Viscosity	The thickness of the sample	Panel definition
Dusty sensation	The feeling of dust or very small particles in the sample	Panel definition
After texture		
Mouth-drying	Drying sensation in the mouth after swallowing	Panel definition
Mouth-watering	Salivation after swallowing	Panel definition
Mouth-coating	Feeling of a coating in the mouth after swallowing the sample	Panel definition
Aftertaste		
Sour aftertaste	Lingering sour taste	Panel definition
Bitter aftertaste	Lingering bitter taste	Panel definition
Sweet aftertaste	Lingering sweet taste	Panel definition

**Table 3 foods-12-02711-t003:** *p*-values and mean scores for the intensities of all attributes where there was a significant effect of fat and/or sweetener. Lower case letters show the fat or sweetener concentrations that resulted in a significantly different mean score for the specific attribute. Lower case letters can only be compared for mean scores for a specific attribute for either fat or sweetener. There was no interaction effect for the shown attributes.

	Cream Aroma	Sweet Taste Intensity	Bitter Taste Intensity	Cocoa Flavor	Chocolate Flavor	Cream Flavor	Fruity and Lactic Flavor	Burnt and Smoky Flavor	Viscosity	Dusty Sensation	Mouth-Watering	Mouth-Drying	Sour Aftertaste	Bitter Aftertaste
*p*-value (fat)	<0.0001	0.037	0.249	0.009	0.652	0.003	0.185	0.021	0.039	0.0003	0.502	0.144	0.430	0.113
*p*-value (sweetener)	0.393	<0.0001	<0.0001	<0.0001	<0.0001	0.005	<0.0001	<0.0001	0.017	<0.0001	0.002	0.004	0.004	<0.0001
Fat														
High fat (4.5%)	6.4 a	8.9 a	7.2	7.8 b	8.5	8.5 a	6.1	5.9 b	9.5 a	5.8 b	8.1	6.3	6.3	6.5
Low fat (0.1%)	4.4 b	8.5 b	7.7	9.1 a	8.2	5.2 b	5.0	6.9 a	7.9 b	7.6 a	7.9	6.9	6.0	7.1
Sweetener														
No sugar (0%)	5.3	2.8 d	10.6 a	10.5 a	6.1 c	5.8 b	4.7 c	9.1 a	7.8 c	8.5 a	6.9 c	8.4 a	6.7 a	9.9 a
Low sugar (2.5%)	5.6	8.0 c	8.2 b	9.1 b	8.0 b	7.0 a	5.4 b	6.3 b	8.5 bc	6.8 b	7.8 b	6.9 b	6.3 ab	7.2 b
High sugar (5%)	5.1	11.2 b	5.3 c	7.4 c	9.3 a	7.4 a	6.0 ab	5.0 c	9.4 a	5.9 bc	8.7 a	5.5 c	5.5 c	5.1 c
Low sugar (2.5%) + acesulfame-K	5.6	12.3 a	5.5 c	6.8 c	10.0 a	7.2 a	6.2 a	5.2 c	9.1 ab	5.5 c	8.6 a	5.5 c	5.9 bc	5.0 c

## Data Availability

The datasets generated for this study are available from the corresponding author, by request.

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
