# Peer review of "Sensory Interactions between Sweetness and Fat in a Chocolate Milk Beverage"

_foods, 2023, doi:10.3390/foods12142711_

Round 1

Reviewer 1 Report

This manuscript is about sensory interactions between sweetness and fat in a chocolate milk beverage. It is interesting and I think a minor revision is needed. You can find my comments in below:

1. The manuscript must be revised grammatically and the English level of it must be improved by a native editor.

2. The authors must re-write the abstract and conclusion sections. I think some sentences are not needed to be in these sections.

3. In line 43, it is better to use ''five'' instead of ''5''.

4. In line 77, please give an example for masked aromas.    

5. In line 91, please give more details about addition of hydrocolloids.

6. In line 117, please give an appropriate reference at the end of paragraph.

7. In line 146, it is better to write ''The samples were prepared'' instead of ''We prepared the samples''.

8. In line 174, the same mistake like number 7 ''We'' is written.

9. In results and discussion sections, it is better to more compare the obtained results with previous researches done on same field. Please do this and add enough references.  

10. Please increase the DPI values of figures. The quality of them is poor. Also, please make the tables more desirable. They are not looking that much good.   

The manuscript must be revised grammatically and the English level of it must be improved by a native editor.

Reviewer 2 Report

Reducing fat and sugar in foods and beverages is increasingly popular for health reasons, but maintaining optimal sensory properties can be challenging. The study explores the sensory interactions between fat and sweetness in a chocolate milk beverage, finding that reducing fat decreases sweetness intensity while reducing sucrose decreases cream, fruity, and lactic flavors, emphasizing the need to consider the effects on non-sugar or non-fat related attributes. I believe that this article deals well with an important topic. However, a minor revision should be performed.

1. Why was the complex system of chocolate milk chosen as the subject of this research?

2. Is there a standard to follow for the specific steps of sensory analysis used in research?

Author Response

Thank you for the comments. Please see the attachment below
